# Coupling aqueous zinc batteries and perovskite solar cells for simultaneous energy harvest, conversion and storage

Peng Chen [1], Tian-Tian Li[1], Yuan-Bo Yang[1], Guo-Ran Li [1] & Xue-Ping Gao [1,2]✉

Simultaneously harvesting, converting and storing solar energy in a single device represents an ideal technological approach for the next generation of power sources. Herein, we propose a device consisting of an integrated carbon-based perovskite solar cell module capable of harvesting solar energy (and converting it into electricity) and a rechargeable aqueous zinc metal cell. The electrochemical energy storage cell utilizes heterostructural $Co_2P$-CoP-$NiCoO_2$ nanometric arrays and zinc metal as the cathode and anode, respectively, and shows a capacity retention of approximately 78% after 25000 cycles at 32 A/g. In particular, the battery cathode and perovskite material of the solar cell are combined in a sandwich joint electrode unit. As a result, the device delivers a specific power of 54 kW/kg and specific energy of 366 Wh/kg at 32 A/g and 2 A/g, respectively. Moreover, benefiting from its narrow voltage range (1.40–1.90 V), the device demonstrates an efficiency of approximately 6%, which is stable for 200 photocharge and discharge cycles.

[1] Institute of New Energy Material Chemistry, School of Materials Science and Engineering, Nankai University, 300350 Tianjin, China. [2] Renewable Energy Conversion and Storage Center, Nankai University, 300350 Tianjin, China. ✉email: xpgao@nankai.edu.cn

Solar-driven self-powered systems could be promising power sources for wearable smart electronics, Internet of Things (IoT) devices and other electrically powered equipment[1,2]. By converting and storing intermittent solar irradiation, a solar rechargeable system (SRS) could improve the practicability of solar energy and fulfil future demands. Traditional SRSs consist of wire-connected independent solar cells and energy storage modules. Such a four-electrode structure is easy to fabricate and efficient but needs additional inactive components that are redundant, which results in increased cost and wasted space[3,4]. A stand-alone two-electrode structure with a bifunctional light-absorbing and the electroactive electrode is compact and attractive; however, its solar energy utilization efficiency and cycling stability are unsatisfactory due to its poor spectrum response, inefficient charge separation and light/chemical corrosion[5–10]. A three-electrode design based on a multifunctional joint electrode could take the merits of different types of SRSs into consideration and exhibit more advantages. By combining solar cells and secondary batteries, such as Li-ion batteries (LIBs)[11,12], lithium-sulfur batteries (LSBs)[13] or other secondary battery systems[14–19], solar rechargeable battery (SRB) systems can achieve an efficient photocharging mode and high specific energy[20,21]; however, they have inferior power performance. Involving high-power electro-chemical energy storage systems, such as aluminium-ion batteries, into SRBs could be an advisable choice, but these systems are expensive and their corrosive electrolyte components are unfavourable[22]. Integrating faradic capacitors with solar cells as solar rechargeable capacitors (SRCs) could also improve the specific power[23–31]; however, the demand on both the specific energy and overall efficiency is always unsatisfactory. Additionally, when aiming at large-scale applications such as IoT devices, cost and safety issues should also be the main considerations. Consequently, the five essential evaluative dimensions for future SRBs should be the following: high specific energy, high specific power, high overall efficiency, high safety and low cost (4H1L). To the best of our knowledge, no reported works have simultaneously satisfied the abovementioned 4H1L features.

Generally, a high specific power originates from both a rapid mass-transfer rate and high electrochemical reaction kinetics[32], while high specific energy is mostly determined by lightweight active materials that undergo multielectron reactions[33]. Specifically, multivalence transition metal oxides, such as spinel $NiCo_2O_4$ or rock-salt $NiCoO_2$, are favourable for enhancing the specific energy[34]. Intermetallic compounds, such as CoP and $Co_2P$ with their high electrical conductivity and catalytic activity, could be helpful to improve the specific power[35]. In addition to the intrinsic nature of electrode materials, nanoscale dimensions are also of great importance. Specially designed heterostructures could combine the merits of different materials to obtain improved performance. Therefore, when combining a high-capacity cathode with a high-energy Zn metal anode, aqueous zinc batteries should exhibit improved energy and power densities[36,37]. In addition, benefiting from cheap and abundant components, such as zinc metal and KOH, as well as non-flammable water solvents, aqueous zinc batteries have great advantages in regard to their low cost and high safety[38–43]. Based on the advantages of aqueous zinc batteries, the overall efficiency of SRBs could be further enhanced by stable and efficient hole-transport-layer-free carbon-based perovskite solar cells[44]; thus, the final device could be an ideal paradigm for SRBs with 4H1L features.

Herein, we propose an integrated solar rechargeable zinc battery (SRZB) with 4H1L features driven by perovskite solar cells. Specifically, a perovskite light absorber, sandwich joint electrode, aqueous alkaline electrolyte and zinc metal are fabricated layer by layer within one structural unit to produce the SRZB. Heterostructural $Co_2P$-CoP-$NiCoO_2$ nanoneedle arrays (NAs) are prepared in situ on one hydrophilic side of the sandwich joint electrode to achieve high specific energy and power. Benefiting from the integrated device structure, specially designed components and narrow-range voltage-matching mechanism, our inexpensive SRZB shows remarkably high specific energy, high specific power, high safety and high overall efficiency. This indicates that the integration of perovskite solar cells and aqueous zinc batteries within one structural unit is a promising attempt to satisfy 4H1L features for future portable power supplies towards the coming epoch of the IoT.

## Results
Herein, the integrated SRZB has a layer-by-layer structure, where the solar energy-conversion unit and energy storage unit are connected into one structural unit via a sandwich joint electrode (Fig. 1). Following the 4H1L principle, we present a brief comparison of various solar rechargeable devices (Supplementary

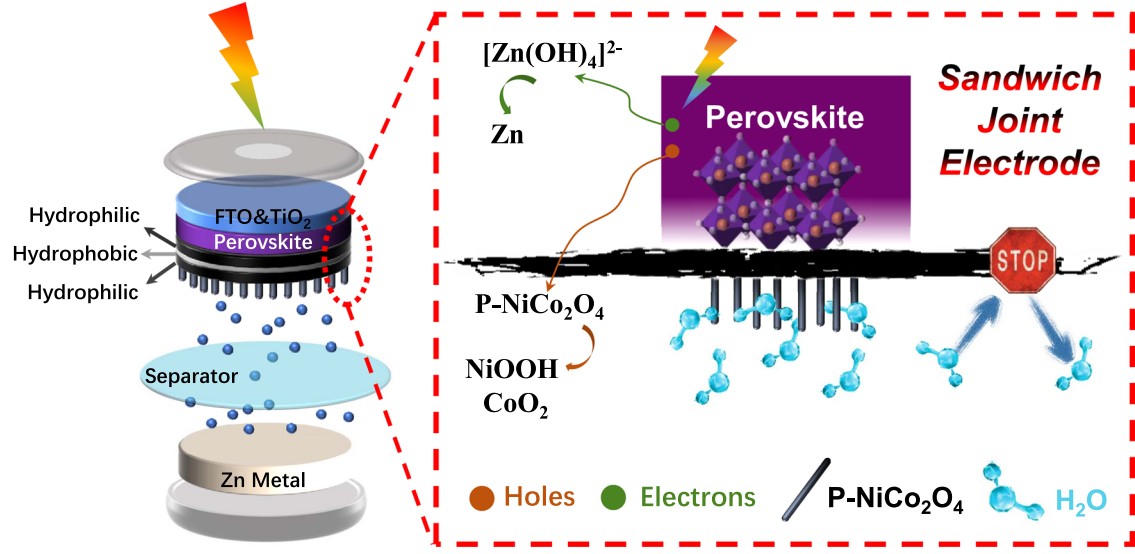

**Fig. 1 Schematic representation of the integrated solar rechargeable zinc battery.** The device consists of a perovskite solar cell part and a rechargeable aqueous zinc metal cell, which are combined via a sandwich joint electrode.

Fig. 1), and SRZB stands out after comprehensive consideration. In particular, the sandwich joint electrode is developed to ensure practicable integration between an aqueous zinc battery and water-sensitive perovskite solar cells to form an SRZB. Intrinsically, the sandwich joint electrode possesses a hydrophilic-hydrophobic-hydrophilic structure feature, which "hides" the impermeable protective layer internally and exposes two external hydrophilic interfaces to simultaneously achieve protection and electrical performance (Fig. 1, Supplementary Figs. 2-4). Electrochemically active P-doped $NiCo_2O_4$ (P-$NiCo_2O_4$) is prepared in situ on one hydrophilic side of the sandwich joint electrode for better aqueous electrolyte compatibility/infiltration, and the other hydrophilic side serves as the counter electrode of perovskite solar cells to obtain better interfacial contact.

Under light illumination, the perovskite layer absorbs photons and produces electron/hole pairs. Then, the photogenerated electrons are injected into $TiO_2$/FTO (fluorine-doped tin oxide) before finally being transferred to the Zn electrode to reduce zinc ions. Moreover, holes spread through the sandwich joint carbon layer to achieve an oxidation reaction on the P-$NiCo_2O_4$-positive electrode[38]. In this way, the SRZB can harness the infinite power of solar irradiation and store this solar energy in terms of electrochemical energy. The as-stored electrochemical energy can be subsequently released with controllable and steady output according to the demand. As a result, intermittent solar irradiation can be collected and stored in situ within SRZB to achieve practicability. During the photocharge and discharge processes, an aqueous electrolyte is trapped within the zinc battery region by the sandwich joint electrode, thus leaving the perovskite solar cell unit undamaged. Generally, the sandwich joint electrode design lays the structural and functional foundation of the integrated SRZB for efficient solar energy conversion and storage.

The active materials on the positive electrode are in situ grown $NiCo_2O_4$ NAs, which are further reacted with $PH_3$ gas to produce P-$NiCo_2O_4$ NAs (Fig. 2a). The $PH_3$ gas flow is generated from the pyrolysis of $NaH_2PO_2 \cdot H_2O$; thus, we can simply control the extent of phosphating by changing the ratio of $NiCo_2O_4$ (NCO) to $NaH_2PO_2 \cdot H_2O$ (P). When the ratio of NCO to P varies from 1:0 to 1:2, the resulting products change from pure spinel NCO to pure rock-salt $NiCoO_2$. After increasing the ratios to 1:4 and 1:8, CoP and $Co_2P$ coexist within the sample (Fig. 2b). During the phosphating process, the morphology of the parent $NiCo_2O_4$ NAs is retained (Fig. 2c and Supplementary Fig. 5), while oxygen is partially replaced by phosphorus, leaving a large number of oxygen vacancies. Additionally, $NiCo_2O_4$ NAs react with $PH_3$ along the axial direction, as shown in Fig. 2a; hence, the resulting $Co_2P$-CoP-$NiCoO_2$ NAs are heterogeneous in the axial direction (Fig. 2d). As demonstrated by EDS mapping in the inset of Fig. 2d, the signal of P in the tail region is visibly less than that in the top and middle areas along the axial direction. Moreover, since the gas-phase reaction occurs mainly on the surface, such a reaction mechanism induces the surface-enriched distribution of phosphides, which is further proven in Fig. 2e, f and Supplementary Fig. 6. As shown, $Co_2P$ nanocrystallines are embedded on the surface of NAs, while a thin CoP layer covers the $NiCoO_2$ porous core. Such a unique heterostructure induces synergy between the photocharge and electrochemical discharge processes. Specifically, the electrochemically active $NiCoO_2$ core result in high specific energy and stability, while CoP and $Co_2P$ nanocrystallines with good electrocatalytic activity on the surface enhance the power output of the device. In contrast, with a higher NCO:P ratio and longer reaction time, NAs are more fragile and easily broken during fabrication and testing, while a low NCO:P ratio leads to low capacity release. Consequently, specially designed $Co_2P$-CoP-$NiCoO_2$ NAs can induce improved energy

and power densities as well as high stability during cyclic electrochemical tests.

As discussed before, the electrochemically active rock-salt $NiCoO_2$ can provide a high capacity and stable structure for long-term electrochemical reactions. With a small amount of P doping (~3% from the SEM and TEM EDS data), the number of $NiCoO_2$ active sites largely increases, as well as the electrical conductivity and catalytic activity; thus, significantly enhanced electrochemical performance is achieved. As shown in Fig. 3a-3c and Supplementary Fig. 7, $Co_2P$-CoP-$NiCoO_2$ NAs exhibit better performance in regard to capacity versus undoped NCO NAs (~210 mAh/g vs. ~30 mAh/g), indicating the pivotal role of P doping and heterostructural design. The coulombic efficiency (CE) and round-trip efficiency increase slightly with a specific current, and the fluctuation at a high specific current is mainly due to the temporal resolution limitation of equipment (Supplementary Fig. 8). The cathode also shows excellent capacity retention at a high specific current (~170 mAh/g retained at 32 A/g, 80.9%), which stems from the synergy of the $Co_2P$-CoP-$NiCoO_2$ heterostructure. Additionally, the long cyclic tests prove that the $Co_2P$-CoP-$NiCoO_2$ cathode works steadily for more than 25000 cycles with a retained capacity of 141 mAh/g (78.3% capacity retention) and CE exceeding 95%. The relatively low and fluctuating coulombic efficiency (>95%) in Fig. 3c is due to side reactions and the inhomogeneous deposition of the Zn anode at a high specific current (32 A/g). When charged/discharged at 2 A/g, the CE exceeds 99% (Supplementary Fig. 9), which can be further improved by a series of surface modifications and electrolyte engineering[45–48]. As a result, the Zn battery unit satisfies the high specific energy, high specific power and stability requirements; moreover, cost-effective components (KOH, Zn, water) and non-flammable aqueous electrolytes enable intrinsically low cost and safety, showing a viable pursuit of 4H1L properties.

When applied as a counter electrode for carbon-based perovskite solar cells (C-PVKs) without a hole-transport layer (HTM), the sandwich joint electrode also presents commendable photovoltaic efficiency and working stability (Fig. 3d, Supplementary Fig. 10). To obtain better stability and a higher specific current, $Cs_{0.15}FA_{0.85}PbI_3$ was chosen as the perovskite absorber. To the best of our knowledge, a power conversion efficiency (PCE) of 14.85% is among the best performing HTM-free C-PVKs[49]. Previously, a Janus joint electrode with hydrophilic and hydrophobic features was used to protect the perovskite layer; however, its use sacrificed smooth surface contact and efficiency[31]. In this work, the contacting surface of the sandwich joint electrode remains hydrophilic; thus, the contact resistance can be minimized, and high efficiency can be retained. Considering the voltage-matching principle of two units, the open-circuit voltage ($V_{oc}$) of the solar cell unit must surpass the redox reaction potential of the corresponding battery unit (Supplementary Fig. 11). Particularly, the maximum power point should be close to the plateau potential to reach high overall efficiency. As a result, we chose a C-PVK module design, which consists of three perovskite solar cells connected in series (Supplementary Fig. 12). The perovskite module shows inferior performance, especially regarding the fill factor, indicating serious charge recombination and high electrical resistance (Fig. 3e). The main reason for the unsatisfactory performance of the C-PVK module is the absence of an HTM layer. However, most HTMs are sensitive to heat, moisture and mechanical damage; thus, the HTM-free C-PVK module structure is chosen to fabricate the integrated SRZB. The band structure of SRZB and the photothermal effect on the aqueous Zn metal cell were measured to probe the charge transfer mechanism involved between the solar absorber and $Co_2P$-CoP-$NiCoO_2$ heterostructure, as well as the structure-function relationship between the solar energy and cathode

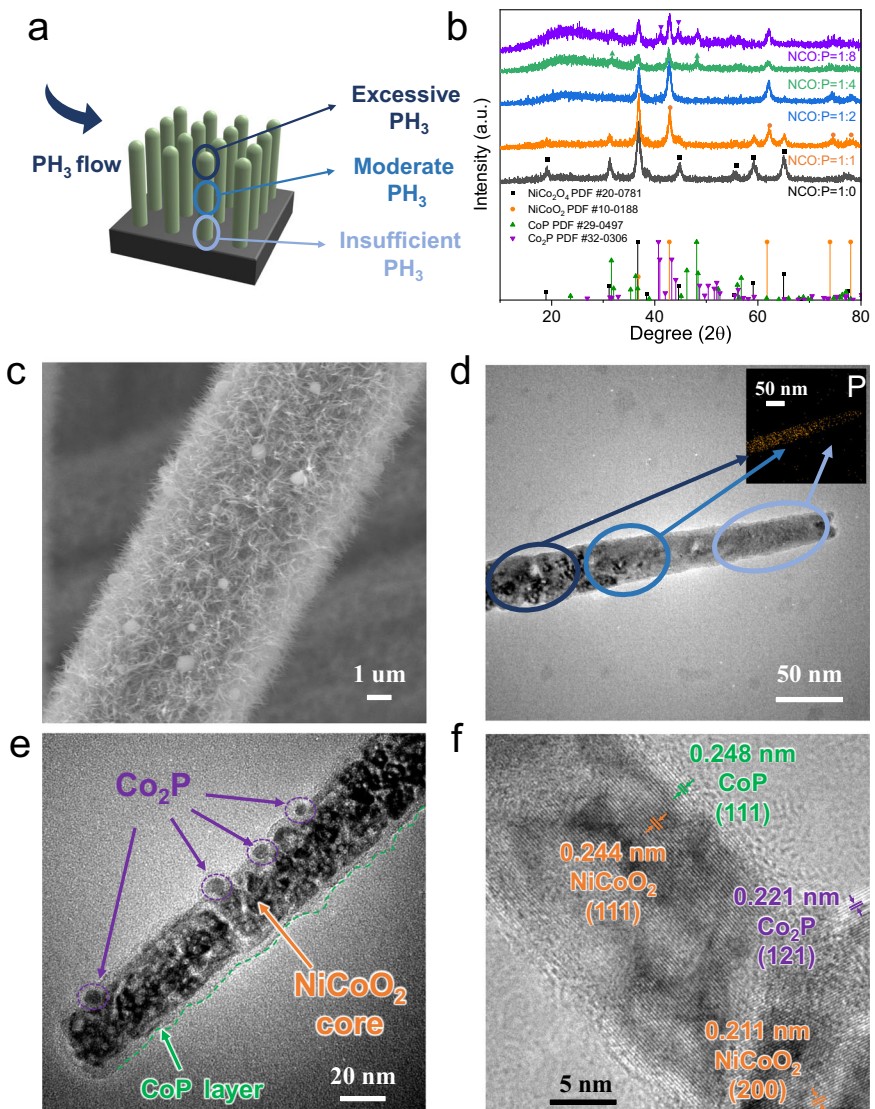

**Fig. 2 Preparation and characterization of P-NiCo₂O₄ NAs. a** Schematic representation of the P-NiCo₂O₄ NA phosphating process. **b** X-ray diffraction (XRD) patterns of the samples with different NiCo₂O₄:NaH₂PO₂·H₂O (NCO:P) ratios. **c** Scanning electron microscopy (SEM) images of P-NiCo₂O₄ NAs on a positive electrode. **d** Transmission electron microscopy (TEM) and energy dispersive spectroscopy (EDS) mapping of Co₂P-CoP-NiCoO₂. **e, f** High-resolution transmission electron microscopy (HRTEM) images of multidimensional heterostructural Co₂P-CoP-NiCoO₂ NAs.

material (Supplementary Figs. 13-15). In general, the working feasibility of SRZB can be ensured by the sandwich joint electrode to provide proper voltage matching between the perovskite module and aqueous Zn cell, thereby making it possible to achieve solar energy conversion and storage with 4H1L features. After assembling the integrated SRZB (Supplementary Fig. 16), the photocharge characteristics of SRZB are measured by using a solar simulator (Zolix, China) and automatic battery tester system (Land, China), as shown in Fig. 4a. Specifically, the device is first activated for several cycles by a land tester. After full discharge, the device is placed under solar irradiation for photocharging, and the charging rate is controlled by modulating the active testing area (i.e. area exposed to solar irradiation) with masks. When the voltage of SRZB reaches 1.9 V during the photocharging process, the galvanostatic discharge process at 8 A/g starts after turning off the light (Fig. 4a). All the photocharge and discharge experiments were controlled manually. Interestingly, the cell shows a good charge efficiency in the relatively narrow voltage range, and ~94.1% of the photocharging process occurs within 1.75–1.90 V (0.21 cm²). Such a narrow voltage-matching

mechanism can ensure higher overall efficiency by achieving a more accurate energy conversion between the two working units near the maximum power point. Moreover, the cut-off voltage of the SRZB (1.9 V) is still far from the $V_{oc}$ (2.4 V) of the C-PVK module, determining a relatively high and steady photocharge current along with the depth of charge. Benefiting from the coupling of the C-PVK module and aqueous Zn metal cell unit, the photocharge rate can be promoted with high-capacity retention. Figure 4b shows the capacities and overall efficiencies of SRZB with different active areas. Clearly, the discharge capacity of SRZB decreases with an increase in the active area, mainly because of the slightly reduced capacity at the faster charge rate in the aqueous Zn metal cell unit (Fig. 3b). Comparatively, the overall efficiency increases to 6% when the active area is increased from 0.21 to 0.42 cm², owing to better current and voltage matching. However, when the active area is further increased, the PCE loss of the C-PVK module dominates the overall efficiency and results in poor overall efficiency (Fig. 4b). In this work, an electrode with an active area of 0.42 cm² was selected for subsequent investigation based on the consideration of high overall

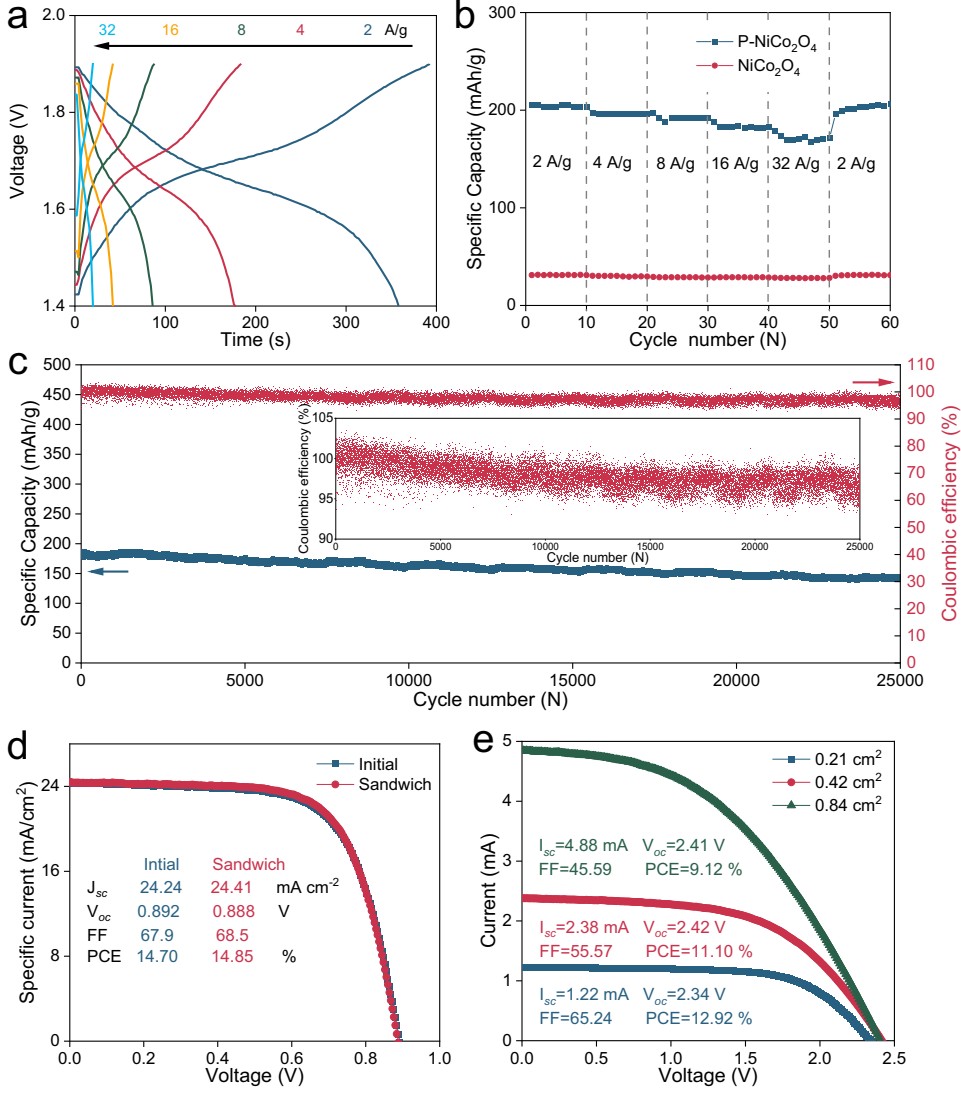

**Fig. 3 Electrochemical performance of aqueous Zn||Co₂P-CoP-NiCoO₂ cells and perovskite solar cells. a** Voltage profiles at various specific currents for the aqueous Zn||Co₂P-CoP-NiCoO₂ cell. **b** Capacity comparison of undoped and P-doped NiCo₂O₄-based Zn batteries at various specific currents. **c** Long-term cycling performance of the aqueous Zn||Co₂P-CoP-NiCoO₂ cell at 32 A/g. **d** J–V profiles of HTM-free C-PVKs based on bare carbon paper and sandwich electrodes. **e** Current–voltage profiles of perovskite modules connected by three HTM-free C-PVKs.

efficiency. When the SRZB is fully charged under light illumination, the as-stored solar energy in the form of chemical energy can be released to electrical energy at different current densities, as shown in Fig. 4c. Owing to the excellent rate capability inherited from the Zn battery unit, the SRZB can maintain a high capacity at a high specific current. In particular, when discharged at 2 A/g, the specific energy of the SRZB can reach 366.4 Wh/kg. Furthermore, high specific power of 54.01 kW/kg is obtained at 32 A/g, while high specific energy can still be maintained at 293.1 Wh/kg (Fig. 4d). The simultaneous photocharging capability of the SRZB was also studied by discharging at different specific currents under light illumination. As shown in Supplementary Fig. 17, the discharge time and capacity increase largely along with a reduced specific current, which can be "infinite", while the discharge current is smaller than the photocharge current. Compared with an independent wire-connected four-electrode structure[3,4], the SRZB shows improved discharge capacity when tested under freezing conditions (0 °C), which benefits from the photothermal effect (Supplementary Figs. 13, 18 and 19). These results indicate that our SRZB holds great

advantages in terms of its concurrently high energy and high power.

Benefiting from the impermeable sandwich joint electrode in the SRZB, the device delivers steady operating performance in the presence of an aqueous electrolyte. As shown in Fig. 5a, the C-PVK module exhibits a slightly decreased PCE, which is mainly related to the destruction caused by long-term light, heat, moisture and oxygen exposure. In particular, the stability of the C-PVK module with a complex structure is slightly lower than that of a single cell (Supplementary Fig. 10). In contrast, the SRZB presents a different trend in terms of overall efficiency. The initial overall efficiency is ~5.7% (at 0.42 cm² for photocharge and 8 A/g for discharge). After 50 activation cycles, the highest overall efficiency can reach 6.4%. Even at the 200th cycle, the overall efficiency remains steady at ~5.9% (Fig. 5a). As shown in Supplementary Fig. 14, the energy level of Co₂P-CoP-NiCoO₂ is suitable for fast hole extraction from the perovskite layer and can remain nearly unchanged during the 200 electrochemical cycles, which is helpful for maintaining the high overall efficiency of the SRZB. Moreover, the photocharge and discharge profiles are

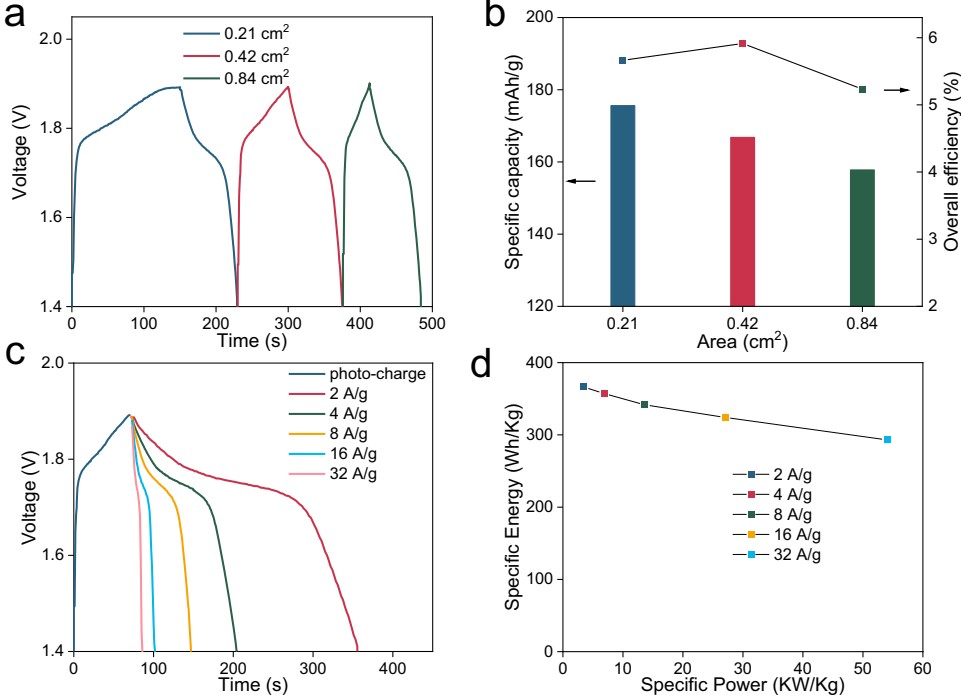

**Fig. 4 Photocharge and galvanostatic discharge profiles of the SRZB. a** Photocharge and galvanostatic discharge profiles of the SRZB with different active areas confined by masks. **b** Capacity and overall efficiency of the SRZB with different active areas confined by masks. **c** Discharge voltage profiles of the SRZB at different specific currents after photocharging. **d** Comparison of the specific power and energy of the SRZB when discharged at various specific currents.

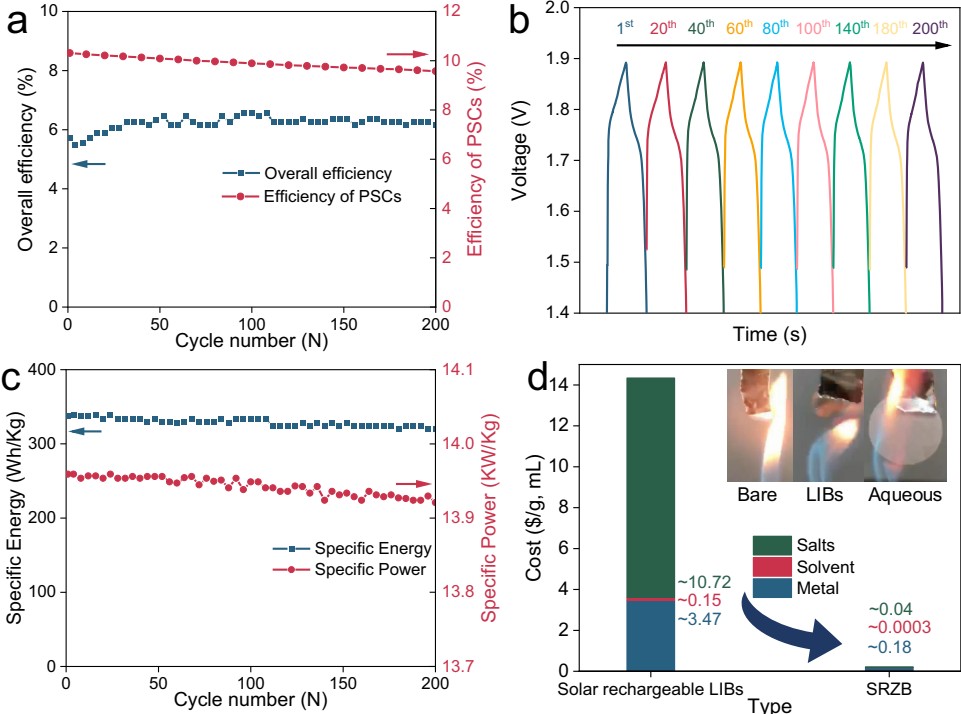

**Fig. 5 Long-term cycling performance and cost analysis of the SRZB. a** Power conversion efficiency of the C-PVK module and overall efficiency of the integrated devices during 200 photocharge and discharge cycles (0.42 cm² and 8 A/g). **b** Voltage-time profiles during 200 photocharge and discharge cycles. **c** Specific energy and power performance during 200 photocharge and discharge cycles. **d** Cost and safety analysis between solar rechargeable LIBs and the SRZB. Insets are digital images of the ignition of the bare, LIB electrolyte-saturated and aqueous electrolyte-saturated separators.

nearly identical for the 200 cycles (Fig. 5b), verifying the stable structure and feasible working mechanism. Additionally, the discharge capacity of the SRZB is relatively stable at ~170 mAh/g over the 200 cycles (Supplementary Fig. 20). Considering that the aqueous Zn metal cell unit can work for more than 25,000 cycles, the SRZB has the potential for long-term use. Figure 5c presents the specific energy and power trends during 200 cycles of photocharge and discharge (at 0.42 cm$^2$ for photocharge and 8 A/g for discharge). Even at the end of the cyclic test, the SRZB can still maintain high specific energy of 320 Wh/kg and high specific power of 13.9 kW/kg. Additionally, the device takes advantage of cost-effective and safe materials, such as water, KOH and zinc metal; thus, the cost of the SRZB is significantly reduced. When compared with solar rechargeable LIBs, the cost of the SRZB in terms of the metal electrode (Li vs. Zn, ~3.47 vs. 0.18 $/g), electrolyte solvent (dry esters vs. water, ~150 vs. 0.3 $/L) and electrolyte salt (dry Li salts, e.g. LiTFSI vs. KOH, ~10.7 vs. 0.04 $/g) are several orders of magnitude lower (Fig. 5d, for details of the costs, see the Method section). Apart from the material cost, the long cycling stability of the SRZB also helps lower the life cycle cost. Additionally, aqueous solutions as electrolytes enable intrinsic safety in terms of thermal runaway when compared with flammable organic electrolytes. After igniting for 8 seconds, the separator saturated with LIB electrolyte burns completely, while the separator saturated with the aqueous solution is still intact (Fig. 5d insets). Moreover, unlike the commonly used power-supply charge mode, in the photocharge process, the charge voltage is determined by the solar conversion unit, which will never surpass $V_{oc}$; thus, theoretically and physically, there should never be overcharging. In this way, the SRZB can satisfy 4H1L features and be a promising candidate for future portable power sources aimed at large-scale outdoor applications.

## Discussion

In summary, in this work, we propose a solar rechargeable zinc battery (SRZB) with high energy, high power, high efficiency, high safety and low-cost (4H1L) features to simultaneously achieve solar energy conversion and storage. Based on heterostructural Co$_2$P-CoP-NiCoO$_2$ nanoneedle arrays (NAs) as the sandwich joint electrode, an integrated battery is fabricated with an aqueous zinc battery unit and an efficient HTM-free C-PVK module. Additionally, the narrow-range voltage-matching mechanism between the zinc battery unit and C-PVK module is achieved, resulting in the high overall efficiency of the SRZB. As a result, the device provides high specific energy of 366 Wh/kg (discharge at 2 A/g), high specific power of 54.01 kW/kg (discharge at 32 A/g), the high overall efficiency of 6.4% and a steady operation for more than 200 cycles with little performance degradation. High safety and low-cost demands are also satisfied owing to the use of cost-effective materials, a safe aqueous electrolyte and a special photocharge mode. Therefore, the integrated SRZB can effectively fulfil the 4H1L principle and shows significant advantages over traditional solar rechargeable devices.

## Methods

**Material preparation**. Etched FTO (fluorine-doped tin oxide) and CH(NH$_2$)$_2$I (FAI) were purchased from Advanced Election Technology Co., Ltd. (Yingkou, China), and PbI$_2$, CsI and CsCl were purchased from p-OLED Corporation (Xi'an, China). Commercial carbon paste was purchased from Shanghai MaterWin New Materials Corporation (China). Toray Carbon Paper was purchased from Canrd (China). Ni(NO$_3$)$_2$·6H$_2$O, Co(NO$_3$)$_2$·6H$_2$O, urea and ethylene glycol were purchased from J&K Chemical Tech (China).

**Sandwich joint electrode fabrication**. We first prepared one-sided polystyrene (PS)-protected carbon papers[25]. A piece of Toray carbon paper was cut into 3 × 5 cm pieces, and after washing with acetone, the carbon paper was placed on a hotplate at 140 °C. A 100 mg/mL polystyrene/chlorobenzene (PS/CB) solution was

gently dropwise added onto the carbon paper on the hotplate to prepare carbon paper with one side protected by PS. The temperature at this stage was important for controlling the one-side deposition of PS. After securing the PS protection, the modified carbon paper was treated with HNO$_3$:H$_2$O (1:4) solution for 30 minutes at 90 °C. After the PS protection process, different synthesis steps occurred. Briefly, 1.5 mmol of Ni(NO$_3$)$_2$·6H$_2$O, 3 mmol of Co(NO$_3$)$_2$·6H$_2$O and 15 mmol of urea were dissolved in 40 mL of distilled water/ethylene glycol mixed solvent (1:1) under vigorous stirring to form a homogeneous solution. After stirring for 0.5 h, the solution was transferred into an autoclave and maintained at 150 °C for 12 h. After the reaction, the NiCo$_2$O$_4$ nanoneedle precursor was deposited in situ on both sides of the carbon paper. After a CB washing process, the PS side was cleared, including the NiCo$_2$O$_4$ precursor on this side. The carbon paper with NiCo$_2$O$_4$ deposited on one side was then obtained by annealing the above product at 300 °C for 8 h in air. After cooling, the carbon paper was placed in a tube furnace with 50 mg of NaH$_2$PO$_2$·H$_2$O and heated to 300 °C in an Ar flow. After phosphating, the black sample turned slightly grey, and the final areal mass loading was ~0.3 mg/cm$^2$. Then, water and ethanol were used to wash the carbon paper several times. Finally, another carbon paper piece and an EVA film were used to glue the two carbon papers together on a hotplate with 1 MPa of pressure for 1 min, and the sandwich joint electrode was obtained after cooling down.

**C-PVK module fabrication**. Etched FTO glasses were cleaned sequentially with detergent, deionized water, alcohol and acetone, followed by drying with a N$_2$ flow and O$_2$ plasma treatment for 5 min. A compact TiO$_2$ layer was deposited by hydrothermal reaction with 40 and 240 mM TiCl$_4$ solution at 75 °C, and a porous TiO$_2$ layer was deposited by spin-coating a diluted solution of TiO$_2$ paste (Greatcell solar Co., Ltd.). The film was annealed at 150 °C for 10 min and 500 °C for 30 min. The FTO/TiO$_2$ substrate was then transferred to a N$_2$-filled glove box. The perovskite layer was deposited on the TiO$_2$ layer by a spin-coating method. In detail, 461 mg of PbI$_2$, 166 mg of FAI, 13 mg of CsI, 16.8 mg of CsCl and 156 mg of dimethyl sulfoxide (DMSO) were mixed in 0.7 mL of N,N-dimethylformamide (DMF). The perovskite precursor solution (50 μL) was dropwise added on the substrate and spin-coated at 1000 rpm for 12 s and 4000 rpm for 25 s with diethyl ether (1 mL) being dripped on the substrates 18 s prior to the end of the process. The substrates were then annealed at 150 °C for 20 minutes to obtain crystalline perovskite films. After cooling, the substrate was fixed with tape on the desk, and the doctor-blading method was used to prepare a thin buffer carbon layer on the top of the perovskite layer. After 5 min of annealing at 150 °C, carbon paste was smeared on one side of the carbon paper or sandwich joint electrode, and then the viscous side was attached on the top of the previous carbon layer. Finally, the C-PVK module was fabricated after another heat treatment step at 120 °C for 15 min.

**Electrolyte preparation**. An aqueous electrolyte was prepared by dissolving KOH (1.68 g) and Zn(Ac)$_2$ (91.7 mg) in distilled water (10 mL)[36–38,40,50–52].

**SRZB assembling**. The clear side of the sandwich joint electrode was pasted on a C-PVK module substrate by using carbon paste. After heating at 120 °C for 15 min, the unit was left to cool to room temperature. The aqueous electrolyte (25 μL) was then added to the P-NiCo$_2$O$_4$ side of the sandwich joint electrode to infiltrate the active materials, after which a 1.5 × 1.5 cm cellulose separator was placed on the top, using a hot-glue gun to fix and seal the separator in case of electrolyte leakage. After the dropwise addition of aqueous electrolyte (~75 μL) to saturate the separator. The Zn plate electrode (99.9% purity, 1.2 × 1.2 cm with a thickness of 0.5 mm) was capped on the top, and a hot-glue gun was used for encapsulation. Finally, another glass substrate was placed on the Zn electrode, and a hot-glue gun was used to seal the device. A Hoffman clip was used to make the whole device more compact and manoeuvrable.

**Measurements**. All the devices were tested under air conditions using a Keithley 2400 source meter, a Newport Oriel Sol2A solar simulator (300 W) and a Zolix solar simulator. We used the 91150 V Reference Cell and Meter (ORIEL instrument) to calibrate the light intensity to 100 mW cm$^{-2}$ before device testing. The PVK performance parameters were obtained from the current–voltage curves of the solar cells under illumination, while a 0.1 cm$^2$ mask was used to measure the power conversion efficiency. The scan direction was set from 1.2 to −0.1 V, with a scan step of 5 mV and dwell time of 1 ms. During photocharging, the light intensity was 68.5 mW cm$^{-2}$ to reduce the temperature increase and electrolyte evaporation loss. All the photocharge and discharge experiments were controlled manually. The lamp was turned on for photocharging, and when the voltage of the device exceeded a certain value, the lamp was covered. The discharge process was controlled by a Land tester according to the program. Electrochemical energy storage tests were performed in a climatic/environmental chamber, and the cell cycling experiments were performed at room temperature at ~25 °C. X-ray photoelectron spectroscopy (XPS) was performed with a Thermo Scientific™ K-Alpha™+ spectrometer equipped with a monochromatic Al Kα X-ray source (1486.6 eV) operating at 100 W. Samples were analysed under vacuum ($P < 10 − 8$ mbar) with pass energy (eV) of 1.0, source energy of He 21.2 eV, and sample bias of −5 V. The electrical resistance test in Supplementary Fig. 2b was measured by a Keithley

2400 source meter with a voltage range of −1 to 1 V, the probes were placed on the same side or different side of the electrode. The water-permeable experiment in Supplementary Fig. 3 was carried out with a gas gun using a 0.4 MPa $N_2$ flow and 1 mL of water, the water flow was sprayed to the electrode, and a receiver was placed behind the electrode. The water-permeable ratio was calculated from the mass change in the receivers by a precise electronic balance. The stationary water infiltration test in Supplementary Fig. 4 was measured by adding 10 mL of water into four glass bottles and covering them with bare carbon paper, Janus carbon paper, sandwich carbon paper and aluminium foil. The bottles were further sealed by a hot-glue gun. Four bottles were placed upside-down on paper for a certain time, and the fluid levels represented the stationary water infiltration properties.

**Calculations**.

(1) The energy-conversion efficiency of the PVKs (PCE):

$$PCE = J_{sc}*V_{oc}*FF/P*100\% \tag{1}$$

where $J_{sc}$, $V_{oc}$, FF and P are the short-circuit current density (mA/cm$^2$), open-circuit voltage (V), fill factor and incident light power density (100 mW/cm$^2$), respectively.

(2) The overall energy-conversion efficiency ($\eta$) for the entire integrated unit is:

$$\eta = E_{discharge}/(P*S*t)*100\% \tag{2}$$

where $E_{discharge}$, P, S and t are the discharge energy of the SRZB (mWh, from the Land machine), light power density, photoactive area of the C-PVK module (cm$^2$) and photocharge time (h), respectively. P and S are constant in the cyclic tests as mentioned before, and the detailed $\eta$, t and $E_{discharge}$ are presented in Supplementary Fig. 21.

(3) The costs of the materials are mainly calculated from open price data on Alfa Aesar and Sigma–Aldrich. Details are shown below:

Li metal: Product No.10767.22, 99.9%, 2250 ¥/100 g, 3.47 $/g. https://www.alfa.com/zh-cn/catalog/010767/

Zn metal: Product No.10437.BW, 99.9%, 780 ¥/665 g, 0.18 $/g. https://www.alfa.com/zh-cn/catalog/010437/

Ethylene carbonate: Product No.676802-6X1L, anhydrous 99%, 5982 ¥/6 L, 0.15 $/mL. https://www.sigmaaldrich.cn/CN/zh/product/sial/676802?context=product

Water: >18 MΩ·cm, produced from water purifier machine, approximately 2 ¥/L, 0.0003 $/mL.

LiTFSI: Product No. H27307.09, 98%, 694 ¥/10 g, 10.7 $/g. https://www.alfa.com/zh-cn/catalog/H27307/

KOH: Product No. A16199.36, 85%, 135 ¥/500 g, 0.04 $/g. https://www.alfa.com/zh-cn/catalog/A16199/

**Reporting summary**. Further information on research design is available in the Nature Research Reporting Summary linked to this article.

## Data availability

The data that support the findings of this study are available from the authors on reasonable request.

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

## Acknowledgements

This work is supported by the National Natural Science Foundation of China (21875123) and the Fundamental Research Funds for the Central Universities, Nankai University (63211043).

## Author contributions

P.C. and X.-P.G. conceived the idea. P.C. carried out the preparation and electrochemical tests of the devices. P.C. and X.-P.G. cowrote the manuscript. T.-T.L. and Y.-B.Y. helped optimize the solar cell performance. G.-R.L. helped analyse the material structure. All the authors contributed to the general discussion.

## Competing interests

The authors declare no competing interests.
