## [Peer Review File · Nature Communications]

REVIEWER COMMENTS

Reviewer #1 (Remarks to the Author):

The authors developed Solar Rechargeable Zinc Battery that can drive Perovskite Solar Cells. The topic of this manuscript is interesting; however, there are some concerns that should be addressed before publication.

1. The introduction of the manuscript is not updated – there are some interesting photo-rechargeable zinc ion battery papers recently published such as *Energy Environ. Sci.*, 2020, 13, 2414-2421; doi.org/10.1002/aenm.202100115; *ACS Energy Lett.* 2020, 5, 3132–3139, *Nano Lett.* 2021, 21, 3527–3532, etc. The authors should clarify how their manuscript is different from these reports as well as advantages.
2. Authors should write “Supplementary Figure S...” instead of “Supplementary Figure ...” throughout the manuscript.
3. There are few Supplementary Figures such as Figure S10, S11, etc. are not at all explained anything in the manuscript.
4. Please explain how authors managed to measure long-term photo charging and discharge test (Figure 5a,5b).
5. The authors provided the overall energy-conversion efficiency formula but not mentioned the used values. For example, values of $E_{\text{discharge}}$, P , S , and t ? – please include them.
6. Authors should include the photo-charge and discharges in dark and light illumination at the same current density to study the simultaneous photo-charging capability of the photo-battery.
7. Please explain why the authors used KOH and $\text{Zn}(\text{Ac})_2$ electrolyte instead of using standard ZnSO_4 electrolyte?

Reviewer #2 (Remarks to the Author):

In this work, the authors proposed an integrated solar rechargeable zinc battery (SRZB) with driven by perovskite solar cells in a single unit where perovskite light absorber, sandwich joint electrode, aqueous alkaline electrolyte and zinc metal are fabricated layer by layer. However, it was not clear to me how it is advantageous over regular solar rechargeable zinc air batteries. If it is an integrated system, any problem in solar cell, it requires to disintegrate the entire unit, however stand-alone set up does not have this disadvantage. Please clarify more on the advantage of integrated SRZB. Other than electrochemical stability, the actual mechanism involved towards the better performance in SRZB, the nature of charge transfer involved between the solar absorber and $\text{Co}_2\text{P-CoP-NiCoO}_2$ hetero-structure is not clear. I request authors to look into the pioneer work in *Nature Communications* volume 10, Article number: 4767 (2019), where solar energy is improving the oxygen evolution reaction kinetics in zinc-air battery. In this work, this is not clear how solar energy will be beneficial for this particular cathode materials. It will be really good to see THE detailed electrochemical studies for the materials for SRZB. What is the round trip efficiency for this battery ? Please comment on this.

Reviewer #3 (Remarks to the Author):

This manuscript presents a successful attempt to couple perovskite solar cell with aqueous zinc battery, with over 6% system efficiency and certain stability. The concept is new and the authors have presented a strategy that overcame certain engineering difficulties. However, the biggest concern is

still the incompatibility of perovskite and aqueous electrolyte. Although the authors have demonstrated the waterproof property of the sandwich electrode for short time, there expected long time light/air exposure will inevitably cause humidity reaction and damage to perovskite. Eventually it might be more cost-effective if the PV and battery modules are independent and wire connected. In my opinion, the (dis)advantage of the integrated device over the one with individual modules must be well clarified, better if the authors can perform parallel experiment on the latter to allow direct comparison. Also a careful proof-reading is required as some typos and awkward sentences have been found. Therefore, I recommend the acceptance of this manuscript after major revision. Below are the additional comments:

1. Page 1-2, "Sunlight is an ideal power source to supply environmental-friendly, cheap and wireless electric energy by photovoltaic technologies", this statement is problematic, consider revise.
2. Page 2, "usher" should be "user"
3. In the introduction, "security" has been mentioned as advantage of Zinc-ion battery; the authors should explain/justify this statement, since "high-capacity cathode with high-energy Zn metal anode", "cheap and abundant ingredients" do not lead to "security".
4. It was not clear how the carbon contact of perovskite cell was made. It was mentioned in the Methods section "carbon paper or carbon paste was attached or doctor-bladed on the top of perovskite layer without hole-transporting-materials", it should be specified exactly which method (paper attached or doctor-bladed) has been used for which figure. How does a carbon paper attached to perovskite can make a device work properly?
5. It would be necessary to mention how the photo-charging and dark-discharging was controlled (Figure 5a). Because it is required to periodically switch on-off the simulator in accordance with the discharging profile, was this done manually or the authors had developed a program to control? What was the duration of the whole test?
6. The authors should present a digital photo of a working device and more clearly describe their fabrication/assembly procedure to allow readers to understand better their work.

List of responses to the reviewer's comments

Reviewer #1:

The authors developed Solar Rechargeable Zinc Battery that can drive Perovskite Solar Cells. The topic of this manuscript is interesting; however, there are some concerns that should be addressed before publication.

*1. The introduction of the manuscript is not updated – there are some interesting photo-rechargeable zinc ion battery papers recently published such as *Energy Environ. Sci.*, 2020, 13, 2414-2421; doi.org/10.1002/aenm.202100115; *ACS Energy Lett.* 2020, 5, 3132–3139, *Nano Lett.* 2021, 21, 3527–3532, etc. The authors should clarify how their manuscript is different from these reports as well as advantages.*

Response: Thanks for your suggestion, we do apologize for missing discussion about those related references. The differences and advantages of this manuscript are mainly proposed from three aspects of structure, mechanism and performance as follows:

Researchers have proposed a series of photo-rechargeable zinc ion battery, lithium ion battery and supercapacitor via a simplified two-electrode design. By using bifunctional electrode materials such as C_3N_4 and V_2O_5 , light energy conversion and chemical/electric energy storage could be achieved within a compact structure. Those devices are free of external solar cells and electronics, in this way, photon absorption, electrons/holes generation/separation, and ion intercalation/deintercalation processes all occur within the cathode.

As a contrast, this manuscript and our past related works are mainly focus on the three-electrode design. Specifically, the solar cells part and secondary battery part are connected by joint electrode within one unit. The joint electrode could act as a counter electrode for a complete solar cell, meanwhile, it could also serve as positive electrode for a complete chemical secondary battery. The solar energy conversion and chemical/electric energy storage processes are operated at the same time but in separated regions. In this manuscript, SRZB could convert solar irradiation into electric energy by perovskite solar cell part and the electric energy would be stored in zinc battery, those two processes are occurred in two parts but simultaneously.

Two-electrode design is usually light, compact and cheap. However, as a compromise, the overall solar energy utilization efficiency (η_{overall}) would be relatively low owing to poor solar spectrum absorption and electron/hole separation efficiency of the bifunctional cathode. The photo to electric efficiency of solar cells could reach 20% and higher at AM 1.5 condition, but most two-electrode photo-rechargeable devices were still suffering the low η_{overall} less than 1%, only very few works could reach 1~2% with sophisticated optimization. As a contrast, three-electrode design could achieve a high and practical η_{overall} of more than 5%, after optimization, the η_{overall} could even reach 10% in specific systems, which is attractive for commercial application. When using Sandwich joint electrode design, our SRZB could be more light, tight, cheap and effective by combining the merits of two types of structure.

To make the introduction clearer, some discussion are added in the manuscript as follows:

Page 2: Traditional SRS consist of wire-connected independent solar cells and energy storage modules, such four-electrode structure is easy-fabricating and efficient but needs extra inactive and repetitive component, thus causing economical and space wasting³⁻⁴. Stand-alone two-electrode structure with bifunctional light absorbing and electroactive electrode is compact and attractive, however, the solar energy utilization efficiency and cyclic stability are unsatisfied due to poor spectrum response, inefficient charge separation, and light/chemical corrosion⁵⁻¹⁰. Three-electrode design based on multifunctional joint electrode could take the merits of different types of SRS into consideration, thus holding more advantages.

Reference:

- 5 Boruah, B. D., Wen, B. & De Volder, M. Light Rechargeable Lithium-Ion Batteries Using V_2O_5 Cathodes. *Nano Lett.* **21**, 3527-3532 (2021).
- 6 Deka Boruah, B. *et al.* Vanadium Dioxide Cathodes for High-Rate Photo-Rechargeable Zinc-Ion Batteries. *Advanced Energy Materials.* **11** (2021).
- 7 Boruah, B. D. *et al.* Photo-rechargeable Zinc-Ion Capacitors using V_2O_5 -Activated Carbon Electrodes. *ACS Energy Letters.* **5**, 3132-3139 (2020).
- 8 Boruah, B. D. *et al.* Photo-Rechargeable Zinc-Ion Capacitor Using 2D Graphitic Carbon Nitride. *Nano Lett.* **20**, 5967-5974 (2020).

9 Boruah, B. D. *et al.* Photo-rechargeable zinc-ion batteries. *Energy & Environmental Science*. **13**, 2414-2421 (2020).

10 Liu, X. *et al.* Utilizing solar energy to improve the oxygen evolution reaction kinetics in zinc-air battery. *Nat Commun*. **10**, 4767 (2019).

2. Authors should write “Supplementary Figure S...” instead of “Supplementary Figure ...” throughout the manuscript.

Response: Thanks for your remind. However, as suggested by the Editor, Supplementary Figure SX should be rewritten as Supplementary Figure X.

3. There are few Supplementary Figures such as Figure S10, S11, etc. are not at all explained anything in the manuscript.

Response: Thanks for your comment. A discussion about all the Supplementary Figures is added in the main text. All the description about supplementary figures are listed below:

Page 4: Following the 4H1L principle, we present a brief comparison of various solar rechargeable devices (Supplementary Figure 1).

Page 4: Intrinsically, the Sandwich joint electrode possesses a hydrophilic-hydrophobic-hydrophilic structure feature, which “hide” the impermeable protective layer internally and exposes two external hydrophilic interfaces in order to achieve protective function and electrical performance simultaneously (Figure 1, Supplementary Figure 2-4).

Page 7: During phosphating process, the morphology of the parent NiCo₂O₄ NAs is retained (Figure 2c and Supplementary Figure 5).

Page 7: Moreover, since the gas-phase reaction occurs mainly on the surface, such reaction mechanism would induce the surface-enriched distribution of phosphides, which is further proven in Figure 2e, 2f and Supplementary Figure 6.

Page 9: As shown in Figure 3a-3c and Supplementary Figure 7, Co₂P-CoP-NiCoO₂ NAs exhibit superior performance on capacity versus un-doped NCO NAs.

Page 9: The coulombic efficiency (CE) and round-trip efficiency could increase slightly along with specific current, the fluctuation under high specific current is mainly due to the limitation of equipment temporal resolution (Supplementary Figure 8).

Page 9: When charged/discharged at 2 A/g, the CE could exceed 99% (Supplementary Figure S9).

Page 9: Sandwich joint electrode also presents commendable photovoltaic efficiency and working stability (Figure 3d, Supplementary Figure 10)

Page 10: The open-circuit voltage (V_{oc}) of solar cell unit must surpass redox reaction potential of corresponding battery unit (Supplementary Figure 11)

Page 10: As a result, we chose a C-PVKs module design, which consists of three perovskite solar cells connected in series (Supplementary Figure 12)

Page 10: The band structure of SRZB and photo-thermal effect on Zn battery was measured to probe the nature of charge transfer involved between the solar absorber and $\text{Co}_2\text{P-CoP-NiCoO}_2$ hetero-structure, as well as the structure-function relationship of solar energy and cathode materials (Supplementary Figure 13-15).

Page 11: After assembling the integrated SRZB (Supplementary Figure 16), the photo-charge characteristics of SRZB is measured by using solar simulator (Zolix, China) and automatic battery tester system (Land, China), as shown in Figure 4a.

Page 12: As shown in Supplementary Figure 17, the discharge time and capacity increase largely along with reduced specific current, which could be “infinite” while the discharge current is smaller than photo-charge current. Compared with independent wire-connected four-electrode structure, SRZB shows improved discharge capacity when tested at freezing condition, which is benefiting from the photo-thermal effect (Supplementary Figure 13, 18, and 19).

Page 14: Meanwhile, the discharge capacity of SRZB is relatively stable at about 170 mAh/g during 200 cycles (Supplementary Figure 20).

Page 18: P and S are constant in cyclic tests as mentioned before, detailed η , t and $E_{\text{discharge}}$ are presented in Supplementary Figure 21.

4. Please explain how authors managed to measure long-term photo charging and discharge test (Figure 5a,5b).

Response: Thanks for the good question. All the photo-charging and discharge experiments were controlled manually. The SRZB was placed under light for photo-charging, the voltage was monitored by Land tester. When voltage exceeded 1.9 V, shield the light and switch Land tester to discharge process. The duration of the whole test is about 8 hours, each photo charge/discharge cycle needs about 150 seconds, 200 cycles need about 8 hours. The photo-charge and discharge cyclic test was conducted manually without interruption, thus we couldn't continue it after 200 cycles without automated program. We have added the details about long-term photo charging and discharge test in the main text and method section as follows:

Page 11: All the photo-charge and discharge experiments were controlled manually.

Page 18, Method section: All the photo-charging and discharge experiments were controlled manually. The lamp was turned on for photo-charging, when the voltage of device exceeds 1.9 V, the lamp should be covered. The discharge process is controlled by Land tester according to the program.

5. The authors provided the overall energy-conversion efficiency formula but not mentioned the used values. For example, values of $E_{\text{discharge}}$, P , S , and t ? – please include them.

Response: Thanks for your suggestion. Detailed value of $E_{\text{discharge}}$, P , S and t are provided in the main text and Supplementary Information as follows:

Calculations section:

Where $E_{\text{discharge}}$, P , S and t are discharge energy of SRZB (mWh, from Land machine), light power density, photoactive area of C-PVKs module (cm^2) and photo-charge time (h), respectively. P and S are constant in cyclic tests as mentioned before, detailed η , t and $E_{\text{discharge}}$ are presented in Supplementary Figure 21.

Supplementary Figure 21. $E_{\text{discharge}}$, t and overall efficiency tendency of SRZB during 200 cycles.

6. Authors should include the photo-charge and discharges in dark and light illumination at the same current density to study the simultaneous photo-charging capability of the photo-battery.

Response: Thanks for the good question! Following your suggestion, we have measured the simultaneous photo-charging capability of SRZB under continuous illumination. With 0.42 cm^2 active area, SRZB was charged to 1.9 V and discharge with different specific current under illumination. The discharge time and specific capacity increased quickly with reduced specific current, the test stopped at 8 A/g because if the discharge current is lower than photo current, the discharge process would be “infinite”. The results and discussion are presented in Supplementary Figure 17 and manuscript as follows:

Page 12: The simultaneous photo-charging capability of SRZB was also studied by discharge at different specific current under light illumination. As shown in Supplementary Figure 17, the discharge time and capacity increase largely along with reduced specific current, which would be “infinite” while the discharge current is smaller than photo-charge current.

Supplementary Figure 17. Simultaneous photo-charging capability of the SRZB.

7. Please explain why the authors used KOH and Zn(Ac)₂ electrolyte instead of using standard ZnSO₄ electrolyte?

Response: Thanks for your question. In this work, we choose the alkaline aqueous zinc battery as energy storage battery, in such systems, KOH and Zn(Ac)₂ aqueous electrolyte are more favorable for high performance. The choice of electrolyte mainly referenced those works:

Method section: Aqueous electrolyte was prepared by dissolving KOH (1.68 g) and Zn(Ac)₂ (91.7 mg) in distilled water (10 mL).^{36-38, 40, 50-52}

Reference:

- 36 Huang, Y. *et al.* Weavable, Conductive Yarn-Based NiCo//Zn Textile Battery with High Energy Density and Rate Capability. *ACS Nano*. **11**, 8953-8961 (2017).
- 37 Wang, X. *et al.* An Aqueous Rechargeable Zn//Co₃O₄ Battery with High Energy Density and Good Cycling Behavior. *Adv Mater*. **28**, 4904-4911 (2016).
- 38 Fang, G., Zhou, J., Pan, A. & Liang, S. Recent Advances in Aqueous Zinc-Ion Batteries. *ACS Energy Letters*. **3**, 2480-2501 (2018).
- 40 Zeng, Y. *et al.* Oxygen-Vacancy and Surface Modulation of Ultrathin Nickel Cobaltite Nanosheets as a High-Energy Cathode for Advanced Zn-Ion Batteries. *Adv Mater*, 1802396 (2018).
- 50 Liu, J. *et al.* A Flexible Quasi-Solid-State Nickel-Zinc Battery with High Energy and Power Densities Based on 3D Electrode Design. *Adv Mater*. **28**, 8732-8739 (2016).
- 51 Teng, C. *et al.* Structural and defect engineering of cobaltic oxide nanoarchitectures as an ultrahigh energy density and super durable cathode for Zn-based batteries. *Chem Sci*. **10**, 7600-7609 (2019).
- 52 Zhou, W. *et al.* A scalable top-down strategy toward practical metrics of Ni-Zn

aqueous batteries with total energy densities of 165 Wh kg^{-1} and 506 Wh L^{-1} .
Energy & Environmental Science. **13**, 4157-4167 (2020).

Reviewer #2:

In this work, the authors proposed an integrated solar rechargeable zinc battery (SRZB) with driven by perovskite solar cells in a single unit where perovskite light absorber, sandwich joint electrode, aqueous alkaline electrolyte and zinc metal are fabricated layer by layer. However, it was not clear to me how it is advantageous over regular solar rechargeable zinc air batteries. If it is an integrated system, any problem in solar cell, it requires to disintegrate the entire unit, however stand-alone set up does not have this disadvantage. Please clarify more on the advantage of integrated SRZB.

Other than electrochemical stability, the actual mechanism involved towards the better performance in SRZB, the nature of charge transfer involved between the solar absorber and Co₂P-CoP-NiCoO₂ hetero-structure is not clear. I request authors to look into the pioneer work in Nature Communications volume 10, Article number: 4767 (2019), where solar energy is improving the oxygen evolution reaction kinetics in zinc-air battery. In this work, this is not clear how solar energy will be beneficial for this particular cathode materials. It will be really good to see THE detailed electrochemical studies for the materials for SRZB.

What is the round trip efficiency for this battery? Please comment on this.

Response: Thanks for your constructive questions and suggestions. The point-to-point responses are presented as shown below:

1. The advantage of integrated SRZB.

Response: The advantages of integrated SRZB are mainly proposed from three aspects of structure, mechanism and performance as follows:

In Nature Communications (2019) 10:4767, the solar rechargeable zinc air battery could convert and store solar energy via a simplified two-electrode stand-alone set up, which involves a bifunctional cathode to absorb photons and enhance the OER kinetics

simultaneously. Such two-electrode stand-alone design could be easier to be repaired when facing some problems, the absence of fragile thin solar cells also helps for improving the stability.

However, in two-electrode stand-alone design, the cathode materials are suffering simultaneous light and chemical corrosion, which may lead to serious surface and structure deterioration. During charge-discharge cycling tests, the interface of electrode/electrolyte would change periodically, which would be harmful for solar absorption and device stability. As for integrated design, the electrochemical reactions are separated from solar harvest parts, which would be pivotal for maintaining the solar energy utilization efficiency. The lifetime of solar cells and battery parts could be optimized respectively with targeted technologies, thus the stability of integrated devices could be promoted to the similar level of individual solar cells and battery systems.

Additionally, as a compromise, the overall solar energy utilization efficiency (η_{overall}) of two-electrode design would be relatively low owing to poor solar spectrum absorption and electron/hole separation efficiency of the bifunctional cathode. Although the photo to electric efficiency of solar cells could reach 20% and higher at AM 1.5 condition, most two-electrode photo-rechargeable devices were still suffering the low η_{overall} less than 1%. As a contrast, three-electrode design could achieve a high and practical η_{overall} of more than 5%, after optimization, the η_{overall} could even reach 10% in specific systems, which is attractive for commercial application.

To make it clearer, additional discussion has been added in the manuscript as follows:

Page 2: Traditional SRS consist of wire-connected independent solar cells and energy storage modules, such four-electrode structure is easy-fabricating and efficient but needs extra inactive and repetitive component, thus causing economical and space wasting^{3,4}. Stand-alone two-electrode structure with bifunctional light absorbing and electroactive electrode is compact and attractive, however, the solar energy utilization efficiency and cyclic stability are unsatisfied due to poor spectrum response, inefficient charge separation, and light/chemical corrosion⁵⁻¹⁰. Three-electrode design based on multifunctional joint electrode could take the merits of different types of SRS into consideration, thus holding more advantages.

2. The nature of charge transfer involved between the solar absorber and Co₂P-CoP-NiCoO₂ hetero-structure and how solar energy will be beneficial for this particular cathode materials.

Response: In Nature Communications (2019) 10:4767, solar irradiation is directly absorbed by metal oxide cathode and thus improving the OER kinetics. However, the situation is quite different in our device, most photons between 300-800 nm are absorbed by perovskite layer, only small amount of low-energy visible light and infrared part would act with cathode. Thus the solar energy would affect the cathode mainly from photothermal effect and would be beneficial for better electrochemical performance especially at low temperature condition.

By testing the Zn battery under dark/light/dark/light and cool at 25 °C/dark with heat at 45 °C, it could be inferred that the photothermal effect is the main reason for performance changes. With light illumination, the temperature rises and the discharge specific capacity increases, but the coulombic efficiency drops serious because of side reactions. When applied in cold condition, the advantage of such photothermal effect would be more attractive.

The charge transfer process in SRZB is discussed in page 4-5, the bandgap structure of P-NCO is suitable for holes transporting, photo induced holes could oxidize P-NCO during photo-charging, while photo induced electrons would reduce Zn²⁺.

Additional measurements about the detailed band structure, photo-thermal effect and electrochemical study are provided to make thorough discussion:

Page 10: The band structure of SRZB and photo-thermal effect on Zn battery was measured to probe the nature of charge transfer involved between the solar absorber and Co₂P-CoP-NiCoO₂ hetero-structure, as well as the structure-function relationship of solar energy and cathode materials (Supplementary Figure 13-15).

Supplementary Figure 13 (a) UV-Vis absorption spectra of SRZB, perovskite and Sandwich joint electrode. (b) Photothermal effect of light illumination.

Supplementary Figure 14 (a-b) Ultraviolet Photoelectron Spectrometer (UPS) spectra of NCO. (c-d) UPS spectra of P-NCO. (e) Band structure of SRZB, the data of perovskite, carbon and Zn is from open literatures.

Supplementary Figure 15 Cyclic performance of Zn battery (32 A/g) with light off, light on, light off, light on & cool treatment and light off & heat treatment.

3. The round-trip efficiency for this battery.

Response: The round-trip efficiency is widely used for evaluating the energy utilization efficiency in large-overpotential systems, like metal-oxygen batteries. In this manuscript, the alkaline aqueous zinc battery owns little overpotential, thus the round-trip efficiency is very close to the coulombic efficiency as shown in Supplementary Figure 8. When fabricating the integrated SRZB, all the energy was acquired from light irradiation, thus we use the overall solar energy utilization efficiency (η_{overall}) for discussion as shown in Figure 5.

Supplementary Figure 8. Coulombic efficiency and round-trip efficiency performance of Zn battery.

Reviewer #3:

This manuscript presents a successful attempt to couple perovskite solar cell with aqueous zinc battery, with over 6% system efficiency and certain stability. The concept is new and the authors have presented a strategy that overcame certain engineering difficulties. However, the biggest concern is still the incompatibility of perovskite and aqueous electrolyte. Although the authors have demonstrated the waterproof property of the sandwich electrode for short time, there expected long time light/air exposure will inevitably cause humidity reaction and damage to perovskite. Eventually it might be more cost-effective if the PV and battery modules are independent and wire connected. In my opinion, the (dis)advantage of the integrated device over the one with individual modules must be well clarified, better if the authors can perform parallel experiment on the latter to allow direct comparison. Also a careful proof-reading is required as some typos and awkward sentences have been found. Therefore, I recommend the acceptance of this manuscript after major revision. Below are the additional comments:

Response: Thanks for your question. Herein we discussed the advantages and disadvantages of integrated device over independent device from three perspectives of structure, mechanism and performance as follows:

Conventional solar rechargeable systems consist of independent solar cell and battery part, which are connected with external circuit. As you have mentioned, the wire-connected structure is more robust and easy-fabricating. However, external connection of separate solar cells and battery needs extra inactive and repetitive component such as current collector and capsulation, thus causing economical and space wasting. In this work, the counter electrode of perovskite and cathode of zinc battery are integrated within one sandwich joint electrode, resulting in volume and weight saving as well as volumetric and gravimetric specific energy improvement.

Additionally, solar cells could only absorb a certain range of solar spectrum (usually below 800 nm for perovskite solar cells), the infrared light which accounts for about 53% of the sunlight energy is almost wasted. In wire-connected structure, infrared light which through the solar cells part would be blocked, as a contrast, those photons with long wavelength could be fully absorbed by bifunctional cathode in integrated structure.

The photothermal effect would be beneficial for better performance especially at low temperature condition. Moreover, with proper electro, thermal and photo active cathode materials, the solar energy could be further utilized, which might be a photochemical approach for breaking the Shockley-Queisser limit of traditional solar cells.

Consequently, the theoretical overall solar energy utilization efficiency (η_{overall}) and electrochemical performance of integrated design could be higher than independent wire-connected structure. Detailed parallel experimental results and additional discussion are presented as follows:

Page 2: Traditional SRS consist of wire-connected independent solar cells and energy storage modules, such four-electrode structure is easy-fabricating and efficient but needs extra inactive and repetitive component, thus causing economical and space wasting^{3,4}. Stand-alone two-electrode structure with bifunctional light absorbing and electroactive electrode is compact and attractive, however, the solar energy utilization efficiency and cyclic stability are unsatisfied due to poor spectrum response, inefficient charge separation, and light/chemical corrosion⁵⁻¹⁰. Three-electrode design based on multifunctional joint electrode could take the merits of different types of SRS into consideration, thus holding more advantages.

Page 12: Compared with independent wire-connected four-electrode structure, SRZB shows improved discharge capacity when tested at freezing condition, which is benefiting from the photo-thermal effect (Supplementary Figure 13, 18, and 19).

Supplementary Figure 13 (a) UV-Vis absorption spectra of SRZB, perovskite and Sandwich joint electrode. (b) Photothermal effect of light illumination.

Supplementary Figure 18. Schematic diagram of integrated and independent solar rechargeable devices.

Supplementary Figure 19. Photo-charge/discharge (8 A/g) performance of independent device and integrated SRZB at 0 °C.

1. Page 1-2, “Sunlight is an ideal power source to supply environmental-friendly, cheap and wireless electric energy by photovoltaic technologies”, this statement is problematic, consider revise.

2. Page 2, “usher” should be “user”

3. In the introduction, “security” has been mentioned as advantage of Zinc-ion battery; the authors should explain/justify this statement, since “high-capacity cathode with high-energy Zn metal anode”, “cheap and abundant ingredients” do not lead to “security”.

Response of Question 1-3: Thanks for your comments! Following your suggestion, Introduction section is revised as follows:

Page 1-2: Solar driven self-powered systems could be promising power sources for wearable smart electronics, Internet of Things (IoT) devices and other electrical powered equipment.^{1,2} By converting and storage intermittent solar irradiation, solar rechargeable system (SRS) could improve the practicability of solar energy and fulfill the future demands.

Page 3: Besides, benefiting from the cheap and abundant ingredients, like zinc metal and KOH, as well as nonflammable water solvent, aqueous zinc battery has great advantage in low cost and high security.³⁸⁻⁴³

4. It was not clear how the carbon contact of perovskite cell was made. It was mentioned in the Methods section “carbon paper or carbon paste was attached or doctor-bladed on the top of perovskite layer without hole-transporting-materials”, it should be specified exactly which method (paper attached or doctor-bladed) has been used for which figure. How does a carbon paper attached to perovskite can make a device work properly?

Response: Thanks for your question, the details about device fabricating are added in Experimental Section as follows:

Page 16-17 Methods section: The substrates were then annealed at 150 °C for 20 minutes to obtain crystalline perovskite films. After cooling down, the substrate was fixed with tape on the desk, and doctor-blading method was used to prepare a thin buffer carbon layer on the top of perovskite layer. After 5 minutes annealing at 150 °C, carbon paste was smeared on one side of the carbon paper or Sandwich joint electrode, then the viscous side was attached on the top of previous carbon layer. Finally, the C-PVKs module was fabricated after another heat treatment step at 120 °C for 15 minutes.

5. It would be necessary to mention how the photo-charging and dark-discharging was controlled (Figure 5a). Because it is required to periodically switch on-off the simulator in accordance with the discharging profile, was this done manually or the authors had developed a program to control? What was the duration of the whole test?

Response: Thanks for your suggestion. All the experiments were controlled manually in this manuscript. The battery tester system and solar simulator are independent, we have tried to design multi-system synchronized linkage controller but not succeed. The duration of the whole test is about 8 hours, each photo charge/discharge cycle needs about 150 seconds, 200 cycles needs about 8 hours. The cyclic test was conducted manually without interruption, thus we couldn't continue it after 200 cycles, if the automated program could be developed, we suppose the long-term stability tests could last for thousands cycles. We have added the details about long-term photo charging and discharge test in the main text and method section as follows:

Page 11: All the photo-charge and discharge experiments were controlled manually.

Page 18 Methods section: All the photo-charging and discharge experiments were controlled manually. The lamp was turned on for photo-charging, when the voltage of device exceeds 1.9 V, the lamp should be covered. The discharge process is controlled by Land tester according to the program.

6. The authors should present a digital photo of a working device and more clearly describe their fabrication/assembly procedure to allow readers to understand better their work.

Response: Thanks for your comment. Following your suggestion, we have added a series of digital photos of the device and detailed description about fabrication procedure in the manuscript and Supplementary information as follows:

Page 17:

SRZB assembling

The clear side of Sandwich joint electrode was pasted on C-PVKs module substrate by using carbon paste, after heating at 120 °C for 15 minutes, the unit was left to cool down

to room temperature. The aqueous electrolyte (25 μL) was then added to the P-NiCo₂O₄ side of Sandwich joint electrode to infiltrate the active materials, after that, placing a piece of 1.5 cm \times 1.5 cm cellulose separator on the top, using a hot-glue gun to fix and seal the separator in case of electrolyte leakage. Adding aqueous electrolyte (about 75 μL) dropwise into saturate the separator. The Zn plate electrode (99.9% purity, 1.2 cm \times 1.2 cm size and 0.5 mm thickness) was capped on the top, using a hot-glue gun for encapsulation, finally, putting another glass substrate on Zn electrode and using hot-glue gun to sealing the device. Hoffman clip was used to make the whole device more compact and maneuverable.

Supplementary Figure 16. (a) Graphical diagram and (b) Digital images of SRZB. All the metal surfaces which may reflect light and influence the results have been covered.

Thank you and look forward to the comments from you and the reviewers.

Sincerely yours,

X. P. Gao

REVIEWER COMMENTS

Reviewer #1 (Remarks to the Author):

The authors have addressed my concerns. I believe this manuscript can be accepted in the present form.

Reviewer #2 (Remarks to the Author):

I have read all the responses against the comments and found them satisfactory. I recommend the publication of this paper in Nature Communications.

Reviewer #4 (Remarks to the Author):

The authors have addressed some of the comments in the revised manuscript. However, there are some issues need to be further clarified before its publication. The holes transfer from a more positive VB to a less positive VB. The discussion of the band structure for charge transfer should be checked carefully. In addition, the band structure of P-NCO is different from NCO. Its effect on the battery performance is suggested to be discussed.

List of responses to the reviewer's comments

Reviewer #1:

The authors have addressed my concerns. I believe this manuscript can be accepted in the present form.

Response: Thank you so much for your reviewing! We deeply appreciate your recognition of our work, we have indeed benefited a lot from your constructive suggestions.

Reviewer #2:

I have read all the responses against the comments and found them satisfactory. I recommend the publication of this paper in Nature Communications.

Response: Thank you so much for your reviewing! We deeply appreciate your recognition of our work, we have indeed benefited a lot from your constructive suggestions.

Reviewer #4:

The authors have addressed some of the comments in the revised manuscript. However, there are some issues need to be further clarified before its publication. The holes transfer from a more positive VB to a less positive VB. The discussion of the band structure for charge transfer should be checked carefully. In addition, the band structure of P-NCO is different from NCO. Its effect on the battery performance is suggested to be discussed..

Response: Thanks for your suggestion! We do apologize for the unclear exhibition of holes transfer process in the schematic diagram. In electrochemical researches, the potential of standard H_2/H^+ (NHE) is commonly used and specified as 0 V, and the holes transfer from a more positive VB to a less positive VB. In the field of solar cells and semiconductor, the position of 0 eV is the vacuum energy level, which is different from electrochemical scale. Despite the distinction in absolute value, the relative position and potential differences of each energy level is the same in electrochemical scale and semiconductor physical scale. Specifically, in our SRZB, the holes transfer from perovskite to carbon and from carbon to P-NCO during photo-charge process.

In our study, the P-NCO presented superior performances over NCO, the improvements of P-NCO originate from a series of complex and synergetic effects, as you suggested, the band structure differences between P-NCO and NCO is one of the critical factors. In Figure 3, we have shown the better electrochemical performance of P-NCO, which is partly benefited from the optimized band structure. During electrochemical tests, the energy level of P-NCO is almost unchanged during first 200 cycles, which means the cyclic photo-charge and discharge performance (Figure 5) was not significantly affected by the energy level changes of P-NCO (Supplementary Figure 14i).

As reported in many published works related to perovskite solar cells, the band structure could influence the charge transfer process from perovskite layer to joint electrode. We have measured the holes extraction rates of NCO and P-NCO by photoluminescence (PL) spectra, the lower PL intensity indicates much faster and efficient holes extraction of P-NCO (Supplementary Figure 14j). During photo-charging process, the better charge transfer from perovskite to P-NCO could reduce the energy loss and improve overall solar energy utilization efficiency of SRZB.

To make the discussion more clear, additional tests results and descriptions are presented in Supplementary Figure 14 and manuscript as follows:

Page 13: As shown in Supplementary Figure 14, the energy level of $\text{Co}_2\text{P-CoP-NiCoO}_2$ is suitable for fast holes extraction from perovskite layer and could remain almost unchanged during the 200 electrochemical cycles, which would be helpful for maintain the high overall efficiency of SRZB.

Supplementary Figure 14 (a-b) Ultraviolet photoelectron spectra (UPS) of NCO. (c-d) UPS of P-NCO. (e-f) UPS of P-NCO after 200 cycles (8A/g). (g-h) UPS of P-NCO after 500 cycles (8A/g). (i) Band structure of SRZB (versus vacuum energy level at 0 eV), the data of perovskite, carbon and Zn is from open literatures. (j) Photoluminescence (PL) spectra of glass/perovskite/NCO and glass/perovskite/P-NCO.

Thank you and look forward to the comments from you and the reviewers.

Sincerely yours,

X. P. Gao